# Association between statin use, atherosclerosis, and mortality in HIV-infected adults

**Binh An P. Phan** [1]*, **Yifei Ma**[1], **Rebecca Scherzer**[2], **Steven G. Deeks**[3], **Priscilla Y. Hsue**[1]

**1** Division of Cardiology, San Francisco General Hospital, Department of Medicine, University of California, San Francisco, California, United States of America, **2** San Francisco Veteran's Affairs Medical Center and Department of Medicine, University of California, San Francisco, California, United States of America, **3** Positive Health Program, San Francisco General Hospital, University of California, San Francisco, California, United States of America

* binhan.phan@ucsf.edu

## Abstract

### Background

While HIV infection is associated with increased cardiovascular risk, benefit from statin is not well established in HIV-infected adults. We assessed whether statins are associated with a decrease in carotid artery intima-media thickness (cIMT) progression and all-cause mortality in HIV-infected adults who are at elevated ASCVD risk and recommended for statins.

### Methods

Carotid IMT was measured at baseline and follow-up in 127 HIV-infected adults who meet ACC/AHA criteria to be on statins. Inverse probability of treatment weighting (IPTW) was used to address selection bias. Multivariable models were used to control for baseline characteristics.

### Results

28 subjects (22%) were on statins and 99 subjects (78%) were not. Mean cIMT at baseline was 1.2 mm (SD = 0.34) in statin users and 1.1 mm (SD = 0.34) in non-users, and the multivariable adjusted difference was 0.05mm (95%CI -0.11, 0.21 p = 0.53). After 3.2 years of follow-up, average cIMT progression was similar in statin users and non-users (0.062mm/yr vs. 0.058 mm/yr) and the multivariable adjusted difference over the study period was 0.004 mm/yr (95% CI -0.018, 0.025, p = 0.74). All-cause mortality appeared higher in non-statin users compared with statin users, but the difference was not significant (adjusted HR = 0.74, 95%CI 0.17–3.29, p = 0.70).

### Conclusion

In a HIV cohort who had elevated ASCVD risk and meet ACC/AHA criteria for statins, treatment with statins was not associated with a reduction in carotid atherosclerosis progression

**Data Availability Statement:** All relevant data are within the manuscript and its Supporting Information files.

**Funding:** This study was funded by the National Institute of Allergy and Infectious Diseases

(K24AI112393 to PYH).The SCOPE cohort is supported by NIAID (K24AI069994 to SGD), the UCSF/Gladstone Institute of Virology & Immunology (P30AI027763 to SGD), the UCSF Clinical and Translational Research Institute Center (UL1RR024131 to SGD), and the CFAR Network of Integrated Systems (R24AI067039 to SGD). The funders had no role in study design, data collection and analysis, decision to publish, or preparation ofthe manuscript. There was no additional external funding received for this study.

**Competing interests:** The authors have declared that no competing interests exist.

or total mortality. Future studies are needed to further explore the impact of statins on cardiovascular risk in the HIV-infected population.

## Introduction

Elevated levels of serum lipoproteins is an important mediator in the pathogenesis of atherosclerosis [1]. Long-term observational studies have demonstrated an association between hypercholesterolemia and increased risk of atherosclerotic cardiovascular disease (ASCVD) [2]. Over the past 30 years, trials of statins have shown that lowering low-density lipoprotein cholesterol (LDL-C) levels reduces atherosclerosis progression and lowers events in populations with elevated CVD risk [3,4]. Based on the vascular and clinical benefits of statins, current cholesterol treatment guidelines from the American College of Cardiology/American Heart Association (ACC/AHA) recommend statins for individuals with established CVD or elevated ASCVD risk [5].

While CVD mortality has improved for the general population, partly due to statins, the HIV-infected population has seen an increase in CVD mortality over the same time period [6]. Beyond traditional CVD risk factors like tobacco use and hypertension, HIV-specific issues such as exposure to antiretroviral therapy, chronic inflammation, and immune activation may contribute to the increased CVD risk observed in HIV-infected adults [7–9]. The ability of statins to alter atherosclerosis and improve cardiac events in HIV-effected adults has not been well established. The aim of our current study was to assess the association between statin use and progression of atherosclerosis by carotid ultrasound and mortality in HIV-infected adults who have increased ASCVD risk and who are recommended to be on statins.

## Materials and methods

### Participants

Study participants were followed at San Francisco General Hospital and San Francisco Veteran's Affairs Medical Center as part of the SCOPE cohort, a longitudinal observational cohort of HIV-infected individuals. The University of California, San Francisco Committee on Human Research approved this study and all participants provided written informed consent (NCT01519141). For all-cause mortality, participants were followed through December 2015 or until the time of death as determined by the National Death Index or SSDI. Two independent physicians adjudicated cardiovascular death using patient ICD-9 codes provided by the National Death Index or Social Security Death Index. To be considered a cardiovascular death, patients were required to have an ICD-9 code related to cardiovascular pathology in $\geq 1$ of the first 3 ICD-9 codes reported on the death document.

### Clinical and laboratory assessment

Interviews and structured questionnaires were given to all participants at the time of enrollment covering socio-demographic characteristics, CVD risk factors, HIV disease history, medications, and health-related behaviors including drug use. Fasting blood work was drawn to measure serum total cholesterol, triglycerides (TG), and high-density lipoprotein cholesterol (HDL-C). Low-density lipoprotein cholesterol (LDL-C) was calculated using Friedewald's formula except for participants with TG $\geq 400$ mg/dL or $< 40$ mg/dL, where it was measured

directly [10]. High-sensitivity C-reactive protein (hs-CRP) levels were measured using a high-sensitivity assay (Dade Behring, Deerfield, IL).

## ASCVD risk assessment and statin recommendation

All participants in the study meet criteria to be on statin therapy based upon the 2018 ACC/AHA cholesterol treatment guidelines [5]. The ACC/AHA cholesterol treatment guidelines recommended statins for participants if they (1) had established ASCVD, (2) were 21 years old or greater with LDL levels ≥ 190 mg/dL, (2) were 40–75 years old with diabetes, (3) were 40–75 years old with ASCVD risk score of ≥ 7.5%—< 20% using the ACC/AHA risk calculator plus risk enhancers such as chronic HIV infection or with ASCVD risk score of >20%. The ACC/AHA risk calculator included the following variables: age, sex, total cholesterol, HDL-C, smoking status, systolic blood pressure, current treatment with blood pressure medications, diabetes, and race (White, Black, other) [11]. Framingham risk score was calculated using the following variables: age, sex, total cholesterol, HDL-C, smoking status, systolic blood pressure, and current treatment with high blood pressure medications [12]. Risk factors included hypertension, smoking, low HDL-C (< 40 mg/dL), family history of premature coronary heart disease (CHD) (CHD in male first-degree relative < 55 years of age; CHD in female first degree-relative < 65 years of age), and age (men ≥ 45 years; women ≥ 55 years).

## Carotid atherosclerosis assessment

Carotid ultrasound was completed at baseline and after a median of 3.2 years of follow-up. Carotid IMT was assessed using the GE Vivid 7 system and a 10-MHz linear array probe. Twelve segments were measured including the near and far wall of the common carotid artery, bifurcation, and internal carotid artery of both the right and left carotid arteries according to the standard protocol from the Atherosclerosis Risk in Communities (ARIC) Study [13] and as previously described from our group [14,15]. All measurements were performed by one experienced technician on digital images using a manual caliper. The technician was blinded with respect to regard to patient history and clinical characteristics. Carotid plaque was defined as a focal area with IMT greater than 1.5 mm in any segment. Repeat scans and measurements were performed with a noted variation coefficient of 3.4% and intra-class correlation coefficient of 0.98.

## Statistical analysis

We summarized continuous variables using mean (± standard deviation (SD)) or median (interquartile range (IQR)), and categorical variables using percentage, stratified by baseline statin use (moderate/high intensity statin use vs. no statin use). We tested differences between statin users and non-users for continuous variables using t-tests for normally distributed variables, Wilcoxon rank-sum tests for non-normally distributed variables, and using chi-squared tests for categorical variables. We used ordinary linear regression model to examine the association between statin use and baseline carotid IMT, and linear mixed models with random intercepts to model the association of statin use with carotid IMT progression. Cox proportional hazard models were constructed to estimate the hazard ratio (HR) and examine the association with all-cause mortality. We presented multivariable models that adjusted for 1) demographic factors only; 2) additionally, viral suppression status, traditional cardiovascular disease risk factors, and HIV-specific risk factors. We presented marginal structural models by applying inverse probability weighting to account for possible confounding by indication, since participants with worse CVD profiles were more likely to receive statin therapy, while participants with lower CVD risk were less likely to receive statin therapy [16]. We further

sought to identify the factors that significantly associated with IMT progression and all-cause mortality in separate models. Proportionality assumption was assured for all mortality models by testing an interaction term between statin use and survival time (in logarithm form). All analyses were performed using SAS 9.4 (SAS Institute, Cary, NC).

## Results

### Participant characteristics

The unweighted baseline characteristics of the 127 HIV-infected participants who meet ACC/AHA criteria to be on statin therapy are shown in Table 1 stratified by whether the participants were moderate/high intensity statin users or non-users. Those HIV-infected participants who were on statins were older than those not on statins (median 57.7 years vs 51.9 years) with more participants on statins being Caucasian and fewer being African-American. Those participants on statins also had longer exposure to protease inhibitors (median 6.0 years vs 2.8 years), lower prevalence of hepatitis C (4% vs 27%), and more evidence of lipodystrophy (82.1% vs 59.6%). The prevalence of family history of CVD and hypertension were higher in those participants on statins. Those on statins had higher hs-CRP levels. There was a higher prevalence of CVD at baseline in subjects on statins (32% vs 7%). After inverse probability of treatment weighting was applied to correct for potential treatment bias, improved balance was achieved in baseline characteristics between the two statin groups except that statin users had a higher rate of aspirin use and lower prevalence of hepatitis C infection (See S1 Table).

### Carotid atherosclerosis and clinical events

Overall, the mean (±SD) of carotid IMT at baseline was 1.12 mm±0.34mm. Participants on statins had greater baseline carotid IMT (1.22±0.34 mm vs 1.09±0.34 mm, p = 0.09). There was no statistically significant difference after demographic adjustment. After additional multivariable adjustment with inverse probability weighting, the difference remained similar (0.052mm, p = 0.53, Table 2). The median duration of follow-up between baseline and last follow-up ultrasound was 3.2 years. At follow-up, the overall mean carotid IMT progression was 0.084 mm/year (±0.097). At follow-up, there was no significant difference in carotid IMT progression between those with or without statins, 0.069 mm/year (±0.059) vs 0.058 mm/year (±0.050), p = 0.71 (Table 1). In a multivariable adjusted model with inverse probability treatment weight, statin use was not associated carotid IMT progression (Table 2). Factors independently associated with faster carotid IMT progression included age (per 10 year increment, 0.024 mm/year, 95% CI 0.013–0.034, p<0.01), BMI (per 5 unit increment, 0.012 mm/year, 95% CI 0.003, 0.022, p = 0.014), and hypertension (0.024 mm/year, 95% CI 0.008–0.039, p = 0.003).

In the average 10.9 years of follow-up in the study, 24 deaths occurred (3 statin users and 21 non-statin users); of these, 2 were CVD related, and 4 were of unknown cause. In a multivariable Cox survival model with inverse probability treatment weightings, statin users had lower all-cause mortality than non-users but the hazard ratio (HR) was not statistically significant (Table 3). Further analysis that controlled for demographics and nadir CD4 levels showed that HIV duration (HR 0.90, 95% CI 0.81, 099, p = 0.038), hepatitis C (HR 4.60, 95% CI 1.40, 14.8, p = 0.012), and hypertension (HR 1.24, 95% CI 1.24, 11.9, p = 0.020) were significant factors of mortality (Table 4). The Cox survival model showed no violation of proportionality assumption. The log-rank test for the Kaplan-Meier survival curves between the statin and non-statin groups was not significant (p = 0.27).

**Table 1. Unweighted baseline characteristics (median, interquartile range) of HIV-infected adults recommended for statins stratified by statin use.**

|  | Statin (-) (n = 99) | Statin(+) (n = 28) | All (n = 127) | P-value |
|---|---|---|---|---|
| Demographic |  |  |  |  |
| Age, years | 51.9 (48.6, 57.3) | 57.7 (50.4, 62.1) | 54 (48.8, 58.8) | 0.038 |
| Race |  |  |  | 0.005 |
| Caucasian | 53% | 89% | 61% | 0.010 |
| African American | 35% | 7% | 29% |  |
| Latino | 6% | 0% | 5% |  |
| Gender |  |  |  | 0.351 |
| Male | 93% | 100% | 94% |  |
| Female | 5% | 0% | 4% |  |
| Transgender (M>F) | 2% | 0% | 2% |  |
| BMI, kg/m$^2$ | 25 (22, 28) | 26 (23, 30) | 25 (23, 28) | 0.500 |
| SBP, mmHg | 125 (119, 135) | 121 (116, 134) | 124 (118, 135) | 0.258 |
| DBP, mmHg | 78 (72, 83) | 77 (70, 81) | 77 (72, 82) | 0.211 |
| HIV factors |  |  |  |  |
| HIV duration, yr | 15 (10, 19) | 16 (14, 19) | 15 (11, 19) | 0.162 |
| Nadir CD4, cells/uL | 1.8 (0.8, 3.0) | 1.5 (0.2, 2.2) | 1.8 (0.6, 3.0) | 0.166 |
| Cur CD4, cells/uL | 4.3 (2.5, 6.6) | 5.0 (3.3, 6.6) | 4.4 (2.8, 6.6) | 0.196 |
| Treated, suppressed | 48% | 79% | 54% | 0.018 |
| PI, yr | 2.8 (0, 5.7) | 6.0 (2.6, 8.7) | 3.3 (0, 6.0) | 0.0004 |
| HAART, yr | 3.8 (0, 6.0) | 5.7 (3.8, 8.3) | 4.3 (0, 6.5) | 0.010 |
| Hepatitis C | 27% | 4% | 22% | 0.008 |
| Lipodystrophy | 60% | 82% | 65% | 0.028 |
| Comorbidities |  |  |  |  |
| Family history | 17% | 36% | 20% | 0.034 |
| Hypertension | 41% | 68% | 48% | 0.013 |
| DM | 11% | 25% | 16% | 0.063 |
| Any smoking | 78% | 64% | 76% | 0.146 |
| Aspirin use | 26% | 64% | 76% | <0.001 |
| Baseline CVD | 7% | 32% | 13% | <0.001 |
| Labs |  |  |  |  |
| Cholesterol, mg/dL | 192 (166, 217) | 200 (156, 231) | 192 (162, 218) | 0.682 |
| LDL-C, mg/dL | 106 (89, 134) | 85 (75, 134) | 104 (82, 134) | 0.161 |
| HDL-C, mg/dL | 39 (30, 48) | 43.5 (31, 54) | 40 (30, 49) | 0.298 |
| Triglyceride, mg/dL | 160 (98, 292) | 157 (126, 363) | 160 (103, 302) | 0.141 |
| hs-CRP, mg/L | 1.9 (0.6, 4.6) | 2.6 (1.2, 7) | 2.0 (0.9, 4.7) | 0.048 |
| Glucose, mg/dL | 89 (79, 99) | 95 (89, 104) | 90 (82, 101) | 0.021 |
| Risk |  |  |  |  |
| Fram risk factors | 3 (2, 3) | 2 (2, 3) | 3 (2, 3) | 0.794 |
| 10-yr Fram risk score | 10 (8, 14) | 10 (5, 13) | 10 (8, 14) | 0.150 |
| 10-yr ASCVD risk score | 10 (9, 13) | 10 (8, 15) | 10 (9, 13) | 0.745 |

BMI = body mass index, SBP = systolic blood pressure, DBP = diastolic blood pressure, PI = protease inhibitor, HAART = highly active anti-retroviral therapy, DM = diabetes mellitus, LDL-C = low density lipoprotein cholesterol, HDL-C = high density lipoprotein cholesterol, hs-CRP = high sensitivity C-reactive protein, Fram = Framingham, ASCVD = atherosclerotic cardiovascular disease.

**Table 2. Association of statin use with baseline carotid IMT and IMT progression.**

| Outcome | Weighted cIMT (mean±SD) | Demographic-adjusted [a] Estimate (95% CI) | Multivariable adjusted [b] Estimate (95% CI) | Marginal structural model [c] Estimate (95% CI) |
|---|---|---|---|---|
| **Baseline cIMT** | | | | |
| Statin non-user | 1.105±0.338 mm | reference | reference | reference |
| Statin user | 1.417±0.355 mm | 0.046 mm | 0.052 mm | 0.204 mm |
| | | (-0.096, 0.188) | (-0.109, 0.212) | (0.043, 0.365) |
| | | p = 0.53 | p = 0.53 | P = 0.015 |
| **cIMT progression** | | | | |
| Statin non-user | 0.059±0.005 mm/year | reference | reference | reference |
| Statin user | 0.069±0.013 mm/year | 0.005 mm/year | 0.004 mm/year | 0.011 mm/year |
| | | (-0.015,0.026) | (-0.018,0.025) | (-0.014,0.035) |
| | | p = 0.61 | p = 0.74 | P = 0.39 |

[a] Demographic adjusted model includes statin use, age, sex, and race.

[b] Multivariable adjusted model includes statin use, age, sex, race, traditional risk factors, and HIV-related risk factors, as listed in covariates section.

[c] Marginal structural model includes all baseline variables in multivariable model b.

## Discussion

While statin therapy has consistently demonstrated benefit in the general population, the vascular and clinical effects of statins in HIV-infected adults with increased ASCVD risk have not been well established. In our study of HIV-infected adults who had elevated ASCVD risk and meet ACC/AHA criteria to be on statins, we demonstrated that statin therapy had no significant association with carotid IMT progression or all-cause mortality. Our findings may have implications for the cardiovascular care of HIV-infected adults who have elevated CVD risk but may not obtain the expected vascular and clinical benefits from statins.

The ability of statins to reduce CV events in the general population at elevated ASCVD risk has been replicated in a number of randomized controlled trials [17]. A meta-analysis of 10 statin trials enrolling more than 70,000 adults demonstrated that statins significantly reduced the relative risk of all-cause mortality by 22%, major coronary events by 30%, and major cerebrovascular events by 19% [17]. The CV benefits of statins have been hypothesized to be partly related to their ability to inhibit atherosclerosis progression. In the METEOR study, statins use was associated with less carotid IMT progression as compared to placebo in subjects with subclinical atherosclerosis (-0.0014, 95% CI -0.0041–0.0014 mm/year vs. 0.013, 95% CI 0.0087–0.017 mm/yr, p<0.001) [18]. Based upon these atherosclerotic and clinical benefits, the 2018 ACC/AHA cholesterol treatment guidelines recommended statins to individuals with

**Table 3. Association of statin use with all-cause mortality.**

| Outcome | Weighted mortality rate Per 100 person years | Demographic-adjusted [a] HR (95% CI) | Multivariable adjusted [b] HR (95% CI) | Marginal structural model [c] HR (95% CI) |
|---|---|---|---|---|
| **All-cause mortality** | | | | |
| Statin non-user | 1.78 (1.03, 2.85) | reference | Reference | reference |
| Statin user | 0.45 (0.01, 2.40) | 0.59 (0.16, 2.11) p = 0.42 | 0.74 (0.17, 3.29) p = 0.70 | 0.34 (0.04, 2.95) p = 0.32 |

[a] Demographic adjusted model includes statin use, age, sex, and race.

[b] Multivariable adjusted model includes statin use, age, sex, race, traditional risk factors, and HIV-related risk factors, as listed in covariates section.

[c] Marginal structural model includes all baseline variables in multivariable model b.

**Table 4. Multivariable modeling of all-cause mortality adjusted by demographics and significant covariates[*].**

|  | Hazard Ratio | 95% CI | P-value |
| --- | --- | --- | --- |
| Statin use | 0.63 | 0.07, 5.14 | 0.668 |
| HIV duration (per year) | 0.90 | 0.81, 0.99 | 0.038 |
| Hepatitis C | 4.60 | 1.40, 14.8 | 0.012 |
| Hypertension | 3.84 | 1.24,11.9 | 0.020 |

[*] Multivariable adjusted model that controls for age, race, gender, and nadir CD4 level.

established CVD or who were at elevated CVD risk [5]. Adults living with HIV are considered to be at enhanced risk for ASCVD. In HIV-infected adults aged 40–85 years with LDL-C between 70 to 189 mg/dL and a 10-year ASCVD risk of 7.5% or higher, statin therapy carries a class IIa recommendation. In our current study, all of our HIV-infected subjects had elevated CV risk based upon traditional risk assessments and meet 2018 ACC/AHA criteria to be on statin therapy. In addition to all subjects having a CV risk-enhancing factor with chronic HIV infection, 48% had a diagnosis of hypertension, 76% had a history of smoking, and more than 70% of the subjects had carotid atherosclerosis at baseline. Unlike non-HIV infected populations with elevated CVD risk, we found no significant association between statin use and reductions in carotid IMT progression or mortality.

Statins have previously failed to show benefit in certain populations whose CV risk may be driven by non-cholesterol related factors [19,20]. In the AURORA Study of 2,776 subjects with end stage renal disease on hemodialysis, statins lowered LDL-C by 43% but had no significant effect on CV outcomes [19]. In the CORONA study of 5,011 subjects with systolic heart failure, statins lowered LDL-C by 45% but did not lower CV events [20]. This lack of benefit in patients with heart failure was also observed in the GISSI-HF trial [21]. Similar to individuals with advanced renal disease or heart failure, adults infected with HIV have non-lipid associated factors that impact their CV risk and may minimize the benefit of statins [22]. While traditional risk factors such as smoking may partly explain the higher burden CVD risk in HIV-infected adults, there are HIV-specific factors related to immune activation and chronic inflammation that are also important contributors to CV risk [15,23–25]. In prior research, we have shown that inflammatory markers such as hs-CRP levels were independent predictors of baseline and progression of atherosclerosis as measured by carotid IMT in HIV-infected adults [15,25]. In our current study, we found that survival was associated with HIV duration and hepatitis C infection; both of which are not affected by statin use.

The effects of statin therapy have been previously studied in the HIV population; though the trials have been small and mainly limited to HIV-infected adults without established CVD or documented elevated ASCVD risk [26–28]. In a cohort of 20 HIV-infected adults, Stein et al. showed that treatment with 40 mg daily of pravastatin lowered LDL-C by 20%, but had no significant effect on flow-mediated vasodilation of the brachial artery [26]. In addition to cholesterol lowering, statin therapy has been demonstrated to decrease some but not all biomarkers of inflammation in HIV subjects [29]. For example, treatment with 10 mg daily of rosuvastatin for 24 weeks significantly lowered lipoprotein-associated phospholipase A2 by 10% as compared to placebo in 147 HIV-infected adults; however there was no impact on hs-CRP or IL-6. Beyond biomarkers, there have been a limited number of trials evaluating the effects of statin on atherosclerosis in the HIV population [30,31]. In a study of 40 HIV-infected adults with evidence of subclinical coronary atherosclerosis by coronary CT angiography, Lo et al. demonstrated that treatment with atorvastatin 40 mg daily for 12 months reduced non-calcified plaque volume and high-risk plaque features but did not significantly reduce hs-CRP

or IL-6 [30]. In the SATURN-HIV study, 147 HIV-infected subjects with LDL-C < 130 mg/dL and who had evidence of increased inflammation by hs-CRP or T-cell activation were randomized to 10 mg of rosuvastatin or placebo [31]. After 96 weeks, statin therapy was associated with 0.019 mm less carotid IMT progression as compared to placebo; which was not modified by inflammatory or immune activation biomarkers. Our study is distinct in that we focused specifically on a cohort of HIV-infected adults with increased ASCVD risk and who meet national guideline criteria to be on statin therapy. In this patient population, we demonstrate no association between statin use and decreased carotid IMT progression or all-cause mortality.

We hypothesize that factors related to chronic inflammation in our HIV-infected population may have impacted the effects of statins. Beyond the primary mechanism of LDL-C reduction, the CV benefit of statins has been hypothesized to be due in part to their anti-inflammatory properties [32]. In the JUPITER study of 17,802 participants without established CVD, LDL-C < 130 mg/dL, and elevated hs-CRP, treatment with rosuvastatin reduced hs-CRP levels by 37% and the primary CV endpoint by 44% as compared to placebo. However, this benefit was reduced in subjects who had residual elevation in hs-CRP [33]. In the PRO-VE-IT-TIMI 22 study, which randomized 3,745 subjects presenting with an acute coronary syndrome to statin therapy, Ridker et al. showed that in subjects who reached an on-treatment LDL-C < 70 mg/dL, those subjects with an elevated hs-CRP $\geq$ 2 mg/L had a higher risk for recurrent CV events as compared to those with a hs-CRP level of < 2 mg/L (age-adjusted event rate per 100/person year of 3.1 vs 2.4, p<0.001). Additionally, residual inflammation with statins has also been associated with less atherosclerosis regression as assessed by coronary intravascular ultrasound [34]. Whether statins will have an impact on chronic inflammation in HIV remains uncertain since statins did not significantly lower inflammatory markers such as hs-CRP or IL-6 in prior trials conducted in HIV-infected adults [29,30]. In our study, baseline median hs-CRP levels were significantly higher in the statin group as compared to the non-statin group (2.6 vs 1.9 mg/L, p = 0.05). Because hs-CRP was measured only once at baseline, we were unable to determine if statin therapy was associated with reductions in inflammation. A greater degree of inflammation at baseline or follow-up may have impacted the ability of statins to reduce carotid atherosclerosis progression or mortality.

There are some important limitations with our study. This was an observational study, and despite our attempts to adjust our analysis for potential bias, there may be unknown confounders not accounted for in our analysis. Because most study subjects were male or Caucasian, our findings may not be generalizable to women or non-Caucasian populations. Additionally, the numbers of deaths overall were small and not all deaths were due to CVD. A dose-response analysis could not be performed due to the overall small number of statin users.

In summary, we found that in a cohort of HIV-infected adults with elevated ASCVD risk and who were recommended to be on statins, treatment with statins was not associated with significant differences in carotid atherosclerosis progression or a reduction in all-cause mortality as compared to those not on statins. Contributions by HIV-related factors such as chronic inflammation may explain the lack of association between statins and clinical benefits. Future studies are needed to further explore the vascular and clinical effects of statins in the HIV-infected population and help identify those HIV-infected adults who may benefit from statins.

## Supporting information

**S1 Table. Weighted baseline characteristics (median, interquartile range) of HIV-infected adults recommended for statins stratified by statin use.**
(DOCX)

## Author Contributions

**Conceptualization:** Binh An P. Phan, Priscilla Y. Hsue.

**Data curation:** Binh An P. Phan, Steven G. Deeks, Priscilla Y. Hsue.

**Formal analysis:** Binh An P. Phan, Yifei Ma, Rebecca Scherzer, Priscilla Y. Hsue.

**Funding acquisition:** Steven G. Deeks, Priscilla Y. Hsue.

**Investigation:** Binh An P. Phan, Steven G. Deeks, Priscilla Y. Hsue.

**Methodology:** Binh An P. Phan, Yifei Ma, Rebecca Scherzer, Steven G. Deeks, Priscilla Y. Hsue.

**Project administration:** Steven G. Deeks, Priscilla Y. Hsue.

**Resources:** Steven G. Deeks, Priscilla Y. Hsue.

**Supervision:** Steven G. Deeks, Priscilla Y. Hsue.

**Writing – original draft:** Binh An P. Phan.

**Writing – review & editing:** Binh An P. Phan, Yifei Ma, Rebecca Scherzer, Steven G. Deeks, Priscilla Y. Hsue.

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
