## [Decision Letter · Decision Letter 0]

30 Mar 2020

PONE-D-20-01850

Association Between Statin Use, Atherosclerosis, and Mortality in HIV-infected Adults

PLOS ONE

Dear Binh An Phan,

Thank you for submitting your manuscript to PLOS ONE. After careful consideration, we feel that it has merit but does not fully meet PLOS ONE’s publication criteria as it currently stands. Therefore, we invite you to submit a revised version of the manuscript that addresses the points raised during the review process.

Please carefully consider and respond to the points I have made below and to the points identified by the reviewer.  I suspect that you cannot do a dose response analysis due to the small number of subjects, but it would be good to at least have a sense of what doses the statin subjects were taking and how much of a reduction in LDL had been achieved. 

We would appreciate receiving your revised manuscript by May 14 2020 11:59PM. To enhance the reproducibility of your results, we recommend that if applicable you deposit your laboratory protocols in protocols.io, where a protocol can be assigned its own identifier (DOI) such that it can be cited independently in the future. For instructions see: http://journals.plos.org/plosone/s/submission-guidelines#loc-laboratory-protocols

We look forward to receiving your revised manuscript.

Kind regards,

James M Wright

Academic Editor

PLOS ONE

Additional Editor Comments:

There are inconsistencies and inaccuracies in the manuscript that need to be corrected. If there are remaining inconsistencies when you resubmit rejection is likely.

In the results: "In a multivariable Cox survival model with inverse probability treatment weightings, statin users had lower all-cause mortality than non-users but the hazard ratio (HR) was not statistically significant (Table 3)."

In the abstract: "All-cause mortality appeared higher in statin users (1.93 per 100 person-years) compared with non-users (1.00 per 100 person-years), but the difference was not significant (adjusted HR=0.74, 95%CI 0.17-3.29, p=0.70)."

Please provide the number of deaths in each group. You report there were 24 overall.

In Table 1 the baseline LDL level in the statin group is wrong. You report it as 85. I suspect it must be 105. Please check all the numbers in the manuscript as we may not have identified all of them.

In the discussion you use the CORONA trial as an example of a trial where statins had no benefit. Please also cite the GISSI-HF trial Lancet 2008, which is a better quality trial showing the same thing.

Your findings and other findings in the HIV population suggest to me that a randomised placebo controlled trial is needed in this population.

Journal Requirements:

"This study was funded in part by the National Institute of Allergy and Infectious Diseases (K24AI112393 to PYH). The SCOPE cohort is supported in part also by NIAID (K24AI069994 to SGD), the UCSF/Gladstone Institute of Virology & Immunology (P30AI027763 to SGD), the UCSF Clinical and Translational Research Institute Center (UL1RR024131 to SGD), and the CFAR Network of Integrated Systems (R24AI067039 to SGD).

Reviewers' comments:

Reviewer's Responses to Questions

**Comments to the Author**

1. Is the manuscript technically sound, and do the data support the conclusions?

Reviewer #1: Partly

2. Has the statistical analysis been performed appropriately and rigorously? 

Reviewer #1: Yes

3. Have the authors made all data underlying the findings in their manuscript fully available?

Reviewer #1: Yes

4. Is the manuscript presented in an intelligible fashion and written in standard English?

Reviewer #1: Yes

5. Review Comments to the Author

Reviewer #1: Interesting paper. The paper addressed a relevant question on whether the use of statins has any benefits in decreasing atherosclerosis progression and lowering mortality in patients with HIV infection. The authors did a good job of driving home the fact that some patients may not obtain the expected clinical benefits from statins. Overall, I think the general approach is robust and the findings credible. Appears well carried out. The paper addressed a relevant question but the clinical message is not clear.

To improve the interpretable of the findings, I ask that the authors provide more information on the following points:

The author mentioned that “All-cause mortality appeared higher in statin users (1.93 per 100 person-years) compared with non-users (1.00 per 100 person-years), but the difference was not significant (adjusted HR=0.74, 95%CI 0.17-3.29, p=0.70)” Then, the authors concluded that “Unlike non-HIV infected populations with elevated CVD risk, we found no significant association between statin use and reductions in carotid IMT progression or mortality”

The risks associated with statins use are important to me. Because statins are recommended for general population with high CVD risk, known risks may lead to minimize its use. Death is an outcome that is important to most patients. I wonder if more attention should be paid to screening for the causes of increased mortality among statin users.

In order to clarify the relationship between statin use and all-cause mortality over the 10.9 years of follow up time, it is necessary to carry out a dose-response analysis, which has not been evaluated in this study but which is extremely recommended.

What is the median time-to-death between the two groups and was the attributable risk constant over the observation period? It would be helpful to view the cox survival curves (time vs outcome variables).

The attributable risk difference is small. This means that the finding is sensitive to biases and confounding. While controlling confounding is important to make exposure groups comparable. I have a concern about differential biases. Observational studies cannot account for any unmeasured factor that might influence the outcome. The authors used multiple adjustment strategies. Although the study did not find any benefits from statin use, the study design does not allow to exclude the possibility that the findings may be subject to treatment-selection biases, time related biases and residual confounders by indications that caused unbalance between the two groups. Survivors who were treated with statins could have a worse risk profile for CVD than untreated survivors determined by the pathology rather than by the drugs themselves need to be considered.

Please describe the duration of the look-back period for each of the study arms and were there meaningful differences?

The patient involvement in this research was appropriate to its objectives and scope. However, I somehow doubt the conclusions would be broadcast as applying specifically to Caucasian males. Perhaps stressing that even more would help.

6. PLOS authors have the option to publish the peer review history of their article (what does this mean?). If published, this will include your full peer review and any attached files.

Reviewer #1: No

---

## [Author Response · Author response to Decision Letter 0]

14 Apr 2020

Re: Response to Reviewers

Manuscript ID: PONE-D-20-01850

Manuscript title: Association Between Statin Use, Atherosclerosis, and Mortality in HIV-infected Adults

April 13, 2020

Dear James M. Wright, Academic Editor, PLOS ONE

On behalf of my co-authors, I thank you for your review of our manuscript and providing us with an opportunity to make revisions. Please find below responses to each of the points raised by the editor and reviewer. Additionally, please find our submitted marked-up copy of our manuscript and a clean unmarked version. 

Thank you for providing us with the opportunity to improve our manuscript and make it suitable for publication in PLOS One.

Sincerely,

Binh An P. Phan, MD

Professor of Medicine

University of California, San Francisco

United States

Points raised in “Additional Editor Comments”:

Point #1: 

Editor comments:

"In a multivariable Cox survival model with inverse probability treatment weightings, statin users had lower all-cause mortality than non-users but the hazard ratio (HR) was not statistically significant (Table 3)." In the abstract: "All-cause mortality appeared higher in statin users (1.93 per 100 person-years) compared with non-users (1.00 per 100 person-years), but the difference was not significant (adjusted HR=0.74, 95%CI 0.17-3.29, p=0.70)."

Author response:

We agree with this comment noting the conflicting description of overall mortality found in the data table of the results section and what is noted in the abstract. We have corrected the comment in the abstract to be consistent with the data that there was a higher all-cause mortality in the non-statin group that was not significant.

We have corrected the abstract to include the following sentence: “All-cause mortality was higher in the non-statin users compared with statin users, but the difference was not significant (adjusted HR=0.74, 95%CI 0.17-3.29, p=0.07).”

Point #2:

Editor comments:

“Please provide the number of deaths in each group. You report there were 24 overall.”

Author response:

We have included the number of deaths in each group in the results section. There were 3 deaths in the statin users group and 21 deaths in the non-statin users group.

Point #3:

Editor comments:

“In Table 1 the baseline LDL level in the statin group is wrong. You report it as 85. I suspect it must be 105. Please check all the numbers in the manuscript as we may not have identified all of them.”

Author response:

The baseline LDL cholesterol levels were reviewed. The reported median LDL level in Table 1 of 85 mg/dL for the statin group is correct. The average LDL value for this group of statin users was 106 mg/dL. The average and median values are very different for this group as compared to the non-statin group since the number of subjects were small and the LDL levels were skewed due to one subject having very abnormal LDL level. For the non-statin users, the median LDL was 106 mg/dL and average LDL was 112 mg/dL. 

Point #4:

Editor comments:

“In the discussion you use the CORONA trial as an example of a trial where statins had no benefit. Please also cite the GISSI-HF trial Lancet 2008, which is a better quality trial showing the same thing.”

Author response:

We agree with this comment and included a sentence noting the similar negative findings from the GISSI-HF and provided a citation.

Point #5:

Editor comments:

“Your findings and other findings in the HIV population suggest to me that a randomized placebo controlled trial is needed in this population.”

Author response:

We thank you for your comment and agree that we need more robust investigation with a randomized controlled trial to help determine the benefits of statin therapy in this increased risk population.

Journal Requirements:

http://www.plosone.org/attachments/PLOSOne_formatting_sample_main_body.pdf

http://www.plosone.org/attachments/PLOSOne_formatting_sample_title_authors_affiliations.pdf

Author response:

We have revised our manuscript to meet the PLOS One’s style requirement per the template instructions.

"This study was funded in part by the National Institute of Allergy and Infectious Diseases (K24AI112393 to PYH). The SCOPE cohort is supported in part also by NIAID (K24AI069994 to SGD), the UCSF/Gladstone Institute of Virology & Immunology (P30AI027763 to SGD), the UCSF Clinical and Translational Research Institute Center (UL1RR024131 to SGD), and the CFAR Network of Integrated Systems (R24AI067039 to SGD).

Author response:

We have included an amended statement at the end of our cover letter that declares all the funding sources of support received during this study and also have included the updated funding statement that there was no additional external funding received. 

Author response:

We have included a Supporting Information and captions at the end of our manuscript and revised our Supporting Information file.

Review Comments to the Author:

Reviewer #1: Interesting paper. The paper addressed a relevant question on whether the use of statins has any benefits in decreasing atherosclerosis progression and lowering mortality in patients with HIV infection. The authors did a good job of driving home the fact that some patients may not obtain the expected clinical benefits from statins. Overall, I think the general approach is robust and the findings credible. Appears well carried out. The paper addressed a relevant question but the clinical message is not clear.

To improve the interpretable of the findings, I ask that the authors provide more information on the following points:

Reviewer comment #1:

“The author mentioned that “All-cause mortality appeared higher in statin users (1.93 per 100 person-years) compared with non-users (1.00 per 100 person-years), but the difference was not significant (adjusted HR=0.74, 95%CI 0.17-3.29, p=0.70)” Then, the authors concluded that “Unlike non-HIV infected populations with elevated CVD risk, we found no significant association between statin use and reductions in carotid IMT progression or mortality”

The risks associated with statins use are important to me. Because statins are recommended for general population with high CVD risk, known risks may lead to minimize its use. Death is an outcome that is important to most patients. I wonder if more attention should be paid to screening for the causes of increased mortality among statin users.”

Author response:

We appreciate the comment made by the reviewer and have clarified this discordant statement on mortality as noted above in the editor’s comments. The non-significant increase in mortality was noted in the non-statin group and not the statin group. 

Reviewer comment #2:

“In order to clarify the relationship between statin use and all-cause mortality over the 10.9 years of follow up time, it is necessary to carry out a dose-response analysis, which has not been evaluated in this study but which is extremely recommended.”

Author response:

We agree with the reviewer that it would be informative to carry out a dose-response analysis. Unfortunately, due to the small sample size of our study, we did not have power to perform a dose-response analysis and therefore completed our main analysis based upon statin versus non-statin users. We have included the following sentence in the discussion section to highlight this limitation: “A dose-response analysis could not be performed due to the overall small number of statin users.”

Reviewer comment #3:

“What is the median time-to-death between the two groups and was the attributable risk constant over the observation period? It would be helpful to view the cox survival curves (time vs outcome variables).”

Author response:

The number of deaths in each group was small, and therefore we determined that analysis of median time to death was not meaningful. The log-rank test for the Kaplan-Meier survival curves between the statin and non-statin groups was not significant with a p-value of 0.27. We have included a graph of the KM survival curve in the response to reviewer file for your review. In the results section of the manuscript, we have included the sentence: “The log-rank test for the Kaplan-Meier survival curves between the statin and non-statin groups was not significant (p=0.27).”

 Reviewer comment #4:

“The attributable risk difference is small. This means that the finding is sensitive to biases and confounding. While controlling confounding is important to make exposure groups comparable. I have a concern about differential biases. Observational studies cannot account for any unmeasured factor that might influence the outcome. The authors used multiple adjustment strategies. Although the study did not find any benefits from statin use, the study design does not allow to exclude the possibility that the findings may be subject to treatment-selection biases, time related biases and residual confounders by indications that caused unbalance between the two groups. Survivors who were treated with statins could have a worse risk profile for CVD than untreated survivors determined by the pathology rather than by the drugs themselves need to be considered.”

Author response: 

We agree with the reviewer that there may be variables that may confound our findings. We used multiple adjustment strategies to minimize bias but understand that other unknown confounders may be present in our observational study. We included this limitation in the discussion of the paragraph. 

Reviewer comment #5:

“Please describe the duration of the look-back period for each of the study arms and were there meaningful differences?”

Author response:

The median (IQR) of duration was 3.1 years (1.6, 5.2) for non-statin users and 4.1 years (2.0, 5.0) for statin users. P-value was 0.57, not statistically significant.

Reviewer comment #6:

“The patient involvement in this research was appropriate to its objectives and scope. However, I somehow doubt the conclusions would be broadcast as applying specifically to Caucasian males. Perhaps stressing that even more would help.”

 Author response:

We agree with the reviewer. Our study cohort was predominantly male (94%) and Caucasian (61%); which is similar to the demographics of other HIV-infected cohorts in our region. Our findings may not be as applicable to to a female or more diverse HIV-infected population. We have noted that as a limitation of our study and revised it to include the predominant Caucasian representation.

---

## [Editor Report · Decision Letter 1]

20 Apr 2020

Association Between Statin Use, Atherosclerosis, and Mortality in HIV-infected Adults

PONE-D-20-01850R1

Dear Binh An Phan,

We are pleased to inform you that your manuscript has been judged scientifically suitable for publication and will be formally accepted for publication once it complies with all outstanding technical requirements.

With kind regards,

James M Wright

Academic Editor

PLOS ONE
---

## [Editor Report · Acceptance letter]

21 Apr 2020

PONE-D-20-01850R1 

Association Between Statin Use, Atherosclerosis, and Mortality in HIV-infected Adults 

Dear Dr. Phan:

I am pleased to inform you that your manuscript has been deemed suitable for publication in PLOS ONE. Congratulations! Your manuscript is now with our production department. 

With kind regards,

on behalf of

Professor James M Wright 

Academic Editor

PLOS ONE